# *One-Pot* Synthesis, *E-/Z*-Equilibrium in Solution of 3-Hetarylaminomethylidenefuran-2(3*H*)-ones and the Way to Selective Synthesis of the *E*-Enamines

**DOI:** 10.3390/molecules28030963

**Published:** 2023-01-18

**Authors:** Alexandra S. Tikhomolova, Vyacheslav S. Grinev, Alevtina Yu. Yegorova

**Affiliations:** 1Institute of Chemistry, N.G. Chernyshevsky Saratov National Research State University, 83 Ulitsa Astrakhanskaya, 410012 Saratov, Russia; 2Institute of Biochemistry and Physiology of Plants and Microorganisms—Subdivision of the Federal State Budgetary Research Institution Saratov Federal Scientific Centre of the Russian Academy of Sciences (IBPPM RAS), 13 Prospekt Entuziastov, 410049 Saratov, Russia

**Keywords:** furan-2(3*H*)-ones, hetarylaminomethylidene derivatives, *one-pot* reactions, selective synthesis, physicochemical methods, spectroscopy

## Abstract

We describe a method to synthesize a new class of hetarylaminomethylidene derivatives of furan-2(3*H*)-ones. The method uses 5-(4-chlorophenyl)furan-2(3*H*)-one, triethyl orthoformate, and heterocyclic amines with different ring sizes and heteroatoms under refluxing in absolute isopropyl alcohol. The obtained enamines exist in an equilibrium of *E*- and *Z*-isomers, whose configurations relative to the double exocyclic C=C bond were confirmed with a set of NMR spectroscopy data. The *E*-/*Z*-equilibrium of the synthesized compounds is affected by the configuration of the intermediate, the volume of its substituents, the site of enolate attack, the presence of intramolecular interactions of amino components, the time of the transformation, the order of mixing of the initial reagents, and the use of polar solvents in the NMR experiment. The advantages of the method are that the reaction time is short, the product yield is high, and product purification is easy.

## 1. Introduction

The chemistry of furan compounds is a promising direction in modern research owing to their unique physicochemical properties. Furans are used widely as the basis for many pharmaceutical and therapeutic agents that are structurally similar to their natural counterparts. Furans are convenient intermediates in the synthesis of drugs, such as furosemide **1**, a popular diuretic, and cefuroxime **2**, a penicillin derivative [1] (Figure 1). Furan compounds are used as anticancer drugs [2,3,4,5] and serve as a framework for the development of new HIV inhibitors [6,7,8], antioxidant [9] and antibacterial [10,11,12] drugs, and agents for the prevention and treatment of diseases associated with estrogen deficiency [13].

Introduction of an aminomethylidene fragment into the *meso-*position of the furan-2(3*H*)-one ring will allow these compounds to act as highly effective “building blocks” in the synthesis of complex hybrid molecules with practically important properties. Owing to the pronounced push–pull character [14,15] of the C=C bond, which is ensured by the carbonyl group and its conjugated amino group, these enamines are electrophilic substrates. The interaction with various nucleophiles falls on the sterically unhindered β-position of the C=C bond, which makes it possible to modify such systems in various directions. This leads to the improvement of methods used to make new heterocyclic structures with different sets of heteroatoms, which are candidates for drugs.

Various methods to make enamine compounds have been proposed. One very promising direction in organic synthesis is the use of the highly reactive triethyl orthoformate [16,17]. Combining ortho esters with active methylene compounds and amines in a *one-pot* interaction yields a wide class of enamines, and the transformations proceed under mild conditions and in a short period. In this way, various biologically active compounds can be made. Noncyclic compounds, the most common of which are cyclohexanedione derivatives [17,18,19,20], can act as carriers of the active methylene component [18,19]. Active methylene heterocyclic compounds [21] with different ring sizes and heterocycles containing a methyl group in the side chain [22,23] are widely used.

Such transformations can be carried out on the basis of 1,2-diamino-4-phenylimidazole and yield practically promising products of various heterocyclic compounds [24,25,26]. There may be additional heterocyclization yielding dihydroimidazoquinolinones in a *one-pot* reaction of 1,2-diamino-4-phenylimidazole with cyclohexanedione and triethyl orthoformate, proceeding at the C-3 position of the imidazole fragment [27].

Compounds of the furan-2(3*H*)-one series, containing a hetarylaminomethylidene fragment, have been poorly studied due to the lack of simple and reliable synthetic methods and of information about the chemical behavior of the furans in various reactions. Therefore, the synthesis of such systems remains a relevant task.

## 2. Results and Discussion

### 2.1. Optimization of the Reaction Conditions

Previously, we have examined the possibility of synthesizing arylaminomethylidene derivatives of furan-2(3*H*)-ones by a three-component reaction. The reaction proceeds in a *one-pot* mode and includes aryl-substituted furan-2(3*H*)-ones **3**, an ortho ester **4** as an electrophilic agent, and aromatic amines **5a–e** bearing electron-withdrawing substituents [28,29] (Figure 1).

We now propose a method for the preparation of 3-hetarylaminomethylidenefuran-2(3*H*)-ones **9a–f** on the basis of the interaction of 5-(4-chlorophenyl)furan-2(3*H*)-one **7** as a representative 5-arylfuran-2(3*H*)-one with triethyl orthoformate **4** and heterocyclic amines **8a–f** (Figure 2).

To find the optimal conditions for the synthesis of hetarylaminomethylidene derivatives of furan-2(3*H*)-ones **9**, we examined the three-component *one-pot* reaction of 5-(4-chlorophenyl)furan-2(3*H*)-one **7** with an excess of triethyl orthoformate **4** and 2-aminopyridine **8a** as a heterocyclic amine. We investigated the effect of various solvents on the transformation time and the product yield. The solvents were polar protic (absolute ethanol and isopropyl alcohol), polar aprotic (1,4-dioxane), and nonpolar (absolute benzene and toluene) (Table 1).

Under reflux conditions, the best results were achieved with the polar protic solvent anhydrous isopropyl alcohol. After pyridine amines had been introduced, the reaction yielded new 5-(4-chlorophenyl)-3-[(R-pyridin-2-ylamino)methylidene]furan-2(3*H*)-ones **9a,b**. When 2-aminopyridine **8a** was used, the reaction rate was high (25 min). When an electron-donating substituent, the hydroxyl group, was introduced into the 2-aminopyridine molecule, the transformation yielded 5-(4-chlorophenyl)-3-{[(3-hydroxypyridine-2-yl)amino]methylidene}furan-2(3*H*)-one **8b** and the reduction increased the reaction time to 5 min. This could be explained by an increase in the nucleophilic properties of the amino group owing to the electron-donating effect of the hydroxyl group. Of note, the transformation proceeded smoothly in other solvents, with yields of 60–70%.

### 2.2. Scheme for the Synthesis of Enamines ***9a–f***

On the basis of the obtained results, we suggest a probable scheme for the formation of transformation products in several directions. Specifically, in pathway A, the reaction proceeds through the formation of ethoxyimine intermediate **10** in situ by nucleophilic addition of amines **8a–c,f** to triethyl orthoformate **4**, which loses two EtOH molecules. Subsequently, 5-(4-chlorophenyl)furan-2(3*H)*-one **7** reacts with imine **10** to form intermediate **11**, after which another EtOH molecule is eliminated to give the desired products **9a–c,f**. Pathway B can serve as an alternative mechanism, in which the initial reaction is that of the 5-(4-chlorophenyl)-furan-2(3*H*)-one molecule **7** with triethyl orthoformate **4** to form ethoxymethylene derivative **12**, which is further converted with amino components **8a–c,f** through intermediate compounds **11** to desired 3-hetarylaminomethylidenefuran-2(3*H*)-ones **9a–c,f** (Figure 3).

All isolated products were newly synthesized and fully characterized with a set of analytical and detailed spectral data, including those by FTIR and NMR spectroscopy.

In the FTIR spectrum of compound **9a**, recorded in a KBr matrix, the bands attributed to the stretching vibrations of the amino group are observed in the region of 3265–3223 cm^−1^. The absorption band of the stretching mode of the lactone carbonyl appears at 1725 cm^−1^, and the stretching vibrations of the conjugated exocyclic C=C double bond appear at 1638 cm^−1^. This confirms the existence of compounds **9a–f** in the aminomethylidene form. An additional confirmation of the existence of **9a** in the aminomethylidene form comes from the NMR spectroscopy data. Two sets of signals in the ^1^H NMR spectra of compounds **9a–c** allow us to interpret them as the existence of an equilibrium of *E-* and *Z*-forms in solution, the appearance of which is due to the push–pull nature of enamines, in which the furan-2(3*H*)-one moiety acts as an acceptor group [28,29]. The duplication of the signals in the ^1^H NMR spectra recorded in DMSO-*d*_6_ is observed for signals of exocyclic protons, singlets of vinyl protons at C-4 of the furan-2(3*H*)-one fragment, and doublets of NH protons for the *E-* and *Z-*isomers (Figure 2).

### 2.3. E/Z-Isomerization Study of Enamines ***9a–f***

With a 2D NOESY experiment, it was possible to determine the configuration of compounds **9a–f** by using 5-(4-chlorophenyl)-3-[(pyridin-2-ylamino)methylidene]furan-2(3*H*)-one **9a** as an example. It was also possible to assign the signals of the protons for *E-* and *Z-*isomers separately. In the 2D NOESY spectra, the cross peaks of the NH group and the vinyl proton of the furan-2(3*H*)-one fragment, observed at 7.10/10.85 ppm, are due to the steric closeness of the protons. The shape of the spectra and the closeness of the protons correspond to the *E*-configuration. The *Z*-isomer showed cross peaks at 6.96/8.50 ppm, related to the proton at C-4 and the proton of the exocyclic C=C bond (Figure 3). Enamine systems **9a–f** are push–pull compounds with a furanone ring as an acceptor, and the amine heterocyclic fragment acts as a donor group. To evaluate the effect of the donor strength of the amine heterocyclic fragment on the *E-/Z-*equilibrium, we consider a set of enamines with the same acceptor group. The position of *E-/Z-*equilibrium is influenced by two factors: the donor strength and the presence of intramolecular interactions (H-bond, steric repulsion of H atoms). In the case of a favorable combination of these factors (i.e., high donor strength and the absence of intramolecular interactions), the push–pull nature is expected to be maximal with an isomer ratio of approximately 1:1. The assignment of the integral intensities of the signals for the NH-group protons in the ^1^H NMR spectra makes it possible to determine the content of the corresponding isomers in equilibrium mixtures: that of the *Z-*isomer, from the signal at 10.36 ppm (0.13H), and that of the *E-*isomer, from the signal at 10.85 ppm (0.87H) for compound **9a**. Figure 2 shows the ratios between the obtained *E-/Z-*isomers of compounds **9a–f**.

The presence of an OH group in the aminopyridine moiety at the C-3 position led to a complete shift of the equilibrium toward the *Z-*form, as compared to **9a**. Product **9b** existed in the DMSO-*d*_6_ solution in the *Z-*form, and the presence of the *E-*isomer was not detected. This fact can be explained by additional intramolecular interactions among the proton of the OH group, the proton of the NH group, and the oxygen atom of the furanone ring; these interactions make the *Z-*isomer more stable. 

The shift of the *E-/Z-*equilibrium toward the *E-*diastereomer and the decrease in the proportion of *Z-*diastereomers in the NMR spectra in DMSO-*d*_6_ is probably due to the ability of DMSO-*d*_6_ to accept hydrogen bonds [30,31]. However, the effect of other nonpolar deuterated solvents (benzene-*d_6_*, acetone-*d_6_*, chloroform-*d*) on the ratio of *Z*- to *E*-hetarylaminomethylidenefuran-2(3*H*)-ones could not be revealed because of their low solubility. It is known that when NMR spectra are recorded in acetone, closely related arylaminomethylidenefuran-2(3*H*)-ones based on aromatic amines lead to a shift in the equilibrium in favor of the *Z-*configuration, which is stabilized owing to the formation of an intramolecular hydrogen bond. This can be explained by the fact that acetone does not have the properties of a strong acceptor; therefore, an increase in the proportion of *Z*-isomers is most probable [29].

### 2.4. Selective Synthesis of Enamines ***9d,e*** in the E-Form

The optimized conditions, scope, and versatility of this protocol allowed us to investigate transformations with various five-membered heterocyclic amines **8c–f**. We chose heterocyclic amines with a reduced donor load. Efforts to carry out transformations under *one-pot* reaction conditions in the case of aminotriazole and aminothiazole failed, which can be explained by the impossibility of the formation of intermediate **10** by way of pathway A (Figure 3). Thus, we propose an alternative route for the reaction with sequential mixing of the initial components **7** and **4**, which allows us to direct the process through the initial in situ formation of the ethoxymethylene derivative of furan-2(3*H*)-one **12**, which then reacts with an amine reagent, added in portions. The transformation involving 2-aminothiazole **8e** proceeded similarly (Figure 4). 

Compounds **9d**,**e** were shown to have the *E*-configuration, which can be explained, on the one hand, by an increase in the rotation barrier around the C=C bond. On the other hand, the observed selectivity may be due to the increase in the reaction time, as compared to the use of other amines (kinetic control of the reaction), which indicates the transformation of the kinetic *Z*-adduct into the thermodynamically more stable *E*-isomer [32].

### 2.5. Enamine Synthesis Based on Diaminoheterocycle

In the case of diaminoheterocycles, the structure of which includes two amino groups differing in their donor properties, the question arises of the direction of the reaction. When such a diamine (specifically, 1,2-diamino-4-phenylimidazole) is used, this one-pot conversion proceeds smoothly, and the yields are high. The use of 1,2-diamino-4-phenylimidazole as an amino component in this interaction is promising because one could expect interactions involving the “hydrazine” amino group (path A) and the primary amino group (path B) (Figure 5).

When running the reaction under conditions similar to those described above, we found that the aminomethylidene derivative furan-2(3*H*)-one **9f** was formed with the participation of the amino group without affecting the “hydrazine” amino group, which was established from the spectral data. We believe that this serves as another example of the newly discovered inverse α-effect [33]. It was shown that **9f** existed in a DMSO-*d_6_* solution as a mixture of *E-* and *Z-*isomers in a ratio close to 1:1, which can be explained by the high donor properties of the “non-hydrazine” amino group and by the absence of intramolecular interactions, in contrast to **9d,e**.

### 2.6. Evaluation of the Barrier Energy for the E-/Z-Transition of Enamines ***9a–f***

We first evaluated the relative energies of the *E-* and *Z-*isomers to determine the most thermodynamically stable form for each heterocyclic enamine **9a–f** both in vacuum and in DMSO solution. The magnitude of the energy barrier was evaluated by molecular modeling of the probable push–pull *E-/Z-*isomerization. The values of the barriers for **9a–e** were between 44.48 and 55.00 kcal/mol in vacuum and between 28.92 and 39.50 kcal/mol in DMSO, with a minimum for **9c**. Enamines **9d** and **9e**, which completely exist in the *E*-form, were found to have 6.1 and 10.5% higher transition barriers with respect to the corresponding mean value for enamines **9a–f** (Table 2).

These data indicate that the reason for the existence of *E-/Z-*isomers in the obtained ratios is hardly their direct transition. The predominance of the *E*-form is probably associated with the structure of intermediate **11**, which is determined by the attack of the enolate (hydroxy furan) form of furanone **7**.

### 2.7. Stereochemical Interpretation of the Synthesis of Enamines ***9a–f***

Because the reaction often results in the predominance of the less thermodynamically stable *E-*isomers, we propose a stereochemical description of the formation mechanism for the more preferable *E*-form. The more stable *E*-form of ethyl *N*-(hetaryl-2-yl)formimidate **10** contains a diastereotopic plane that can be attacked by the nucleophilic enolate of **7** from different sites (Figure 6).

The attack from where the less sterically bulky hydrogen atom is located is more preferable and gives rise to 5-(4-chlorophenyl)-3-[ethoxy(hetaryl-2-ylamino)methyl]furan-2(3*H*)-one **11**. As a result of the elimination of the ethanol from the antiperiplanar position (if seen along the C–C bond), an *E*-enamine appears. 

Alternatively, the nucleophile attack from the site where the bulkier ethoxy group is located gives rise to isomeric 5-(4-chlorophenyl)-3-[ethoxy(hetaryl-2-ylamino)methyl]furan-2(3*H*)-ones **11′**. The rotation around the simple C–N bond in these structures can stabilize the **11′** form through an intermolecular hydrogen bond. The antiperiplanar elimination of an ethanol molecule gives rise to a Z-form of enamines **9a–f**.

Thus, on the one hand, the formation of *E-*form enamines **9a–f** is preferred over that of the corresponding *Z-*isomers. On the other hand, the content of *Z-*isomers in the product mixture may strongly increase under favorable factors, such as the formation of an intramolecular hydrogen bond and the steric volume of the amine’s heterocyclic part. 

## 3. Materials and Methods

### 3.1. Physical Measurements

FTIR spectra were collected on an FSM-1201 Fourier spectrometer (Infraspek, St. Petersburg, Russia) in the range 4000–400 cm^−1^ with a spectral resolution of 4 cm^−1^. Samples were mixed with ground KBr (FTIR grade, Sigma–Aldrich, Saint Louis, MO, USA) and pressed into pellets by removing water and air traces under reduced pressure. ^1^H (400 MHz) and ^13^C NMR (100 MHz) spectra in DMSO-*d*_6_ were recorded with a Varian (Agilent) 400 spectrometer (Agilent Technologies, Santa Clara, CA, USA), and the internal standard was TMS (see Appendix A). Chemical shifts (*δ*) are reported in ppm. Elemental analysis was done on an Elementar Vario MICRO cube CHNS analyzer (Elementar Analysensysteme GmbH, Hanau, Germany). Melting points were determined on a Stuart*™* SMP10 melting point apparatus (Cole-Parmer, Beacon Road, Stone, Staffordshire, ST15 OSA, UK). The progress of the reaction and the purity of the synthesized compounds were monitored by TLC on ALUGRAM^®^ SIL G UV_254_ plates (MACHEREY-NAGEL GmbH & Co. KG, Düren, Germany), with hexane–ethyl acetate–acetone (2:2:1) as the eluent.

### 3.2. Synthesis and Characterization of Compounds ***9a–c,f***

A mixture of 1 mmol of 5-(4-chlorophenyl)furan-2(3*H*)-one (**7**), 9 mmol of triethyl orthoformate (**4**), and 1 mmol of the corresponding heterocyclic amine (**8a–d**) was refluxed in 7 mL of absolute isopropyl alcohol. The precipitated crystals were filtered, washed with isopropyl alcohol, recrystallized from DMF, and dried.

*(Z,E)-5-(4-chlorophenyl)-3-[(pyridin-2-ylamino)methylidene]furan-2(3H)-one* (**9a**). Yellow crystals (DMF), yield 0.22 g (75%), mp 275–276 °C; FTIR (KBr), ν, cm^−1^: 3265–3223 (NH), 1722 (C=O), 1637 (C=C); ^1^H NMR (400 MHz, DMSO-*d*_6_): *δ* 6.96 (s, 0.13H, Fu), 7.06–7.06 (m, 2H, Py), 7.10 (s, 0.87H, Fu), 7.49 (d, *J* = 8.0 Hz, 2H, Ar), 7.56 (d, *J* = 8.0 Hz, 2H, Ar), 7.75–7.79 (t, 1H, Py), 8.30 (d, *J* = 4.0 Hz, 1H, Py), 8.35 (d, *J* = 12.0 Hz, 0.87H, =CH), 8.54 (d, *J* = 12.0 Hz, 0.13H, =CH), 10.35 (d, *J* = 12.0 Hz, 0.13H, NH), 10.86 (d, *J* = 12.0 Hz, 0.87H, NH), ^13^C NMR (100 MHz, DMSO-*d*_6_): *δ* 100.83 (4-Fu), 110.00 (3-Fu), 112.06 (Py), 119.49 (Py), 125.34, 125.70, 125.75, 128.28, 129.59, 133.27, 135.12 (6-Fu), 139.42 (Py), 146.85 (5-Fu), 148.85 (Py), 151.76 (Py), 170.16 (C=O). Anal. calcd. for C_16_H_11_ClN_2_O_2_: C: 64.33%; H: 3.71%; N: 9.38%; Cl: 11.87%; Found: C: 64.75%; H: 3.98%; N: 9.82%; Cl: 12.06%. 

*(Z)-5-(4-Chlorophenyl)-3-{[(3-hydroxypyridin-2-yl)amino]methylidene}furan-2(3H)-one* (**9b**). Yellow crystals (DMF), yield 0.25 g (80%), mp 258–260 °C; FTIR (KBr), ν, cm^−1^: 3348 (OH), 3289 (NH), 1774 (C=O), 1689 (C=C); ^1^H NMR (400 MHz, DMSO-*d*_6_): *δ* 6.99 (t, *J* = 8.0 Hz, 1H, Py), 7.02 (s, 1H, Fu), 7.24 (d, *J* = 8.0 Hz, 1H, Py) 7.53 (d, *J* = 8.6 Hz, 2H, Ar), 7.60 (d, *J* = 8.7 Hz, 2H, Ar), 7.82 (d, *J* = 4.8 Hz, 1H, Py) 8.57 (d, *J* = 12.7 Hz, 1H, =CH), 10.39 (d, *J* = 12.8 Hz, 1H, NH), 10.92 (s, 1H, OH). ^13^C NMR (100 MHz, DMSO-*d*_6_): *δ* 101.2 (4-Fu), 104.2 (3-Fu), 120.9, 121.3, 121.7, 122.2, 126.0, 128.3, 132.6, 137.4 (6-Fu), 138.7 (Py), 141.0 (C-OH), 146.2, 155.2 (Py), 170.1 (C=O). Anal. calcd. for C_16_H_11_ClN_2_O_3_: C: 61.06%; H: 3.52%; N: 8.90%; Cl: 11.26%; Found: C: 61.55%; H: 3.98%; N: 9.15%; Cl: 11.78%.

*(Z,E)-4-{[(5-[4-chlorophenyl]-2-oxofuran-3[2H]-ylidene)methyl]amino}-1,5-dimethyl-2-phenyl-1,2-dihydro-3H-pyrazole-3-*one (**9c**). Yellow crystals (DMF), yield 0.25 g (81%), mp 229–239 °C; FTIR (KBr), ν, cm^−1^: 3414 (NH), 1751 (C=O), 1685 (C=C), 1649 (C=O); ^1^H NMR (400 MHz, DMSO-*d*_6_): *δ* 2.31 (d, *J* = 8.0 Hz, 3H, CH_3_), 3.05 (d, *J* = 4.0 Hz, 3H, CH_3_), 6.86 (s, 0.14H, Fu), 7.10 (s, 0.86H, Fu), 7.33-7.56 (m, 4H, Ar), 7.88 (s, *J* = 12.0 Hz, 0.86H, =CH), 8.06 (d, *J* = 16.0 Hz, 0.14H, =CH), 9.54 (d, *J* = 12.0 Hz, 0.14H, NH), 9.69 (d, *J* = 12.0, 0.86H Hz, NH); ^13^C NMR (100 MHz, DMSO-*d*_6_): *δ* 10.79 (CH_3_), 36.40 (CH_3_-N), 101.13 (4-Fu), 111.10 (3-Fu), 124.48, 124.99, 125.23, 127.26, 128.82, 129.92, 129.49, 129.64, 129.67, 132.44, 134.96, 143.22 (6-Fu), 144.48, 146.62 (5-Fu), 160.31 (C=O), 170.03 (C=O). Anal. calcd. for C_22_H_18_ClN_3_O_3_: C: 64.79%; H: 4.45%; N: 10.30%; Cl: 8.69%; Found: C: 65.08%; H: 4.13%; N: 9.98%; Cl: 9.13%.

*(Z,E)-5-(4-chlorophenyl)-3-{[(1-amino-4-phenyl-1H-imidazol-2-yl)amino]methylene}furan-2(3H)-one* (**9f**). Yellow crystals (DMF), yield 0.16 g (59%), mp 246–248 °C; FTIR (KBr), ν, cm^−1^: 3327 (NH_2_), 3267 (NH), 1713 (C=O), 1653 (C=C); ^1^H NMR (400 MHz, DMSO-*d*_6_): *δ* 6.04 (s, 2H, NH_2_), 6.14 (s, 2H, NH_2_), 7.00 (s, 0.54H, Fu), 7.20 (s, 0.46H, Fu), 7.16 (s, 0.46H, CH_imidazole_), 7.18 (s, 0.54H, CH_imidazole_), 8.03 (s, 0.54H, =CH), 8.24 (s, 0.46H, =CH), 10.40 (s, 0.46H, NH), 10.69 (s, 0.54H, NH); ^13^C NMR (100 MHz, DMSO-*d*_6_): *δ* 100.98 (4-Fu), 103.63, 109.39 (3-Fu), 116.45 (C_imidazole_), 124.34, 125.42, 125.66, 128.39, 128.79, 129.37, 129.54, 134.89, 138.27 (6-Fu), 145.31 (5-Fu), 156.71, 169.44 (C=O). Anal. calcd. for C_20_H_15_ClN_4_O_2_: C: 63.41%; H: 3.99%; N: 14.79%; Cl: 9.36%; Found: C: 63.86%; H: 4.13%; N: 14.58%; Cl: 9.13%.

### 3.3. Synthesis and Characterization of Compounds ***9d,e***

A mixture of 1 mmol of 5-(4-chlorophenyl)furan-2(3*H*)-one (**7**) and 9 mmol of triethyl orthoformate (**4**) was heated for 30 min in 7 mL of absolute isopropyl alcohol. Next, the corresponding heterocyclic amine (**8d,e**) was added portionwise for 1 h. The precipitated crystals were filtered, washed with isopropyl alcohol, recrystallized from DMF, and dried. 

*(E)-5-(4-chlorophenyl)-3-[(1H-1,2,4-triazol-3-amino)methylidene]-3H-furan-2-*one (**9d**). Brown crystals (DMF), yield 0.19 g (65%), mp 255–256 °C; FTIR (KBr), ν, cm^−1^: 3435 (NH), 3346 (NH), 1738 (C=O), 1650 (C=C); ^1^H NMR (400 MHz, DMSO-*d*_6_): *δ* 7.17 (s, 1H, Fu), 7.46-7.54 (m, 4H, Ar), 7.92 (d, *J* = 12.0 Hz, 1H, =CH), 8.46 (s, 1H, CH_triazole_), 11.31 (d, 1H, *J* = 12.0 Hz, NH)**,** 13.82 (s, 1H, NH_triazole_); ^13^C NMR (100 MHz, DMSO-*d*_6_): *δ* 100.92 (4-Fu), 113.74 (3-Fu), 125.64, 127.78, 128.35, 129.57, 130.27, 133.15, 137.53 (6-Fu), 144.26, 146.16 (5-Fu), 159.05 (C_triazole_), 170.14 (C=O). Anal. calcd. for C_13_H_9_ClN_4_O_2_: C: 54.09%; H: 3.14%; N: 19.41%; Cl: 12.28%; Found: C: 53.92%; H: 3.24%; N: 19.61%; Cl: 11.93%.

*(E)-5-(4-chlorophenyl)-3-[(thiazol-2-ylamino)methylidene]furan-2(3H)-one* (**9e**). Brown crystals (DMF), yield 0.18 g (60%), mp 256–257 °C; FTIR (KBr), ν, cm^−1^: 3265 (NH), 1726 (C=O), 1662 (C=C); ^1^H NMR (400 MHz, DMSO-*d*_6_): *δ* 7.05 (s, 1H, Fu), 7.24 (d, *J* = 4.0 Hz, 1H, =CH_thiazole_), 7.42 (d, *J* = 4.0 Hz, 1H, =CH_thiazole_), 7.51 (d, *J* = 8.0 Hz, 2H, Ar), 7.58 (d, *J* = 8.0 Hz, 2H, Ar), 7.90 (s, 1H, =CH), 11.84 (s, 1H, NH); ^13^C NMR (100 MHz, DMSO-*d*_6_): *δ* 100.42 (4-Fu), 103.48 (3-Fu), 113.99 (C_thiazole_), 125.86, 126.00, 127.96, 129.43, 129.62, 133.68, 134.94 (6-Fu), 140.00 (C_thiazole_), 147.49 (5-Fu), 162.12 (C_thiazole_), 169.78 (C=O). Anal. calcd. for C_14_H_9_ClN_2_O_2_S: C: 55.18%; H: 2.98%; N: 9.19%; Cl: 11.73%; S: 10.52%; Found: C: 54.99%; H: 3.16%; N: 9.50%; Cl: 12.02%; S: 10.87%. 

### 3.4. DFT Calculations

All structures (*E-* and *Z-*forms) were fully optimized by using Becke’s three-parameter hybrid functional combined with the Lee–Yang–Parr correlation functional (B3LYP [34,35,36]) with the 6-311++G(d) basis set. Two stages were used; stage 1 was *in vacuo,* and stage 2 was in a DMSO solution. In the latter case, a polarizable continuum model (PCM) was used. 

To evaluate the magnitude of the energy of the possible *E-/Z-*transitions, we used a formalism proposed for rotation around the partly delocalized C=C bond [29,37]. The optimized coordinates of the *E-*forms were used for relaxed scans of the C(=O)–C=C–N torsion angle with an increment of 2°. The energy profiles of the rotation processes were analyzed for extrema to evaluate the barrier energy. Calculations were also done at two stages—in vacuo and in a DMSO solution (PCM approach). The structure for each of enamines **9a–f** at the global maxima on the curve was optimized as a transition state (**TS**). The Hamiltonians of the **TS**s were analyzed for imaginary vibrational modes.

## 4. Conclusions

We have shown that it is possible to synthesize enamines based on furan-2(3*H*)-one. The synthesized compounds exist in solution as a mixture of *E-* and *Z-*isomers, the ratio between which is determined by a set of factors. One of these factors is the reaction mode. In the *one-pot* mode, the *E-*form predominates, and the sequential mixing of the starting reagents in the case of aminotriazole and aminothiazole yields only the *E-*form, which can also be explained by the higher energy barrier values for the *E-/Z-*transition, as compared with those for other amines. Stereochemically, the synthesis of enamines proceeds preferably through the initial interaction of the triethyl orthoformate and the amine reagent with the formation of an ethoxyimine intermediate. The configuration of the final product, enamine, is determined by the attack of the enolized form of furanone. The proposed mechanism is the most preferable for the formation of *E-*form enamines.

## Data Availability

Not applicable.

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
