# Peer review of "One-Pot Synthesis, E-/Z-Equilibrium in Solution of 3-Hetarylaminomethylidenefuran-2(3H)-ones and the Way to Selective Synthesis of the E-Enamines"

_molecules, 2023, doi:10.3390/molecules28030963_

Round 1
Reviewer 1 Report
This manuscript describes a three-component condensation which provides enamines attached to a furanone moiety. The innovation of this part of the series is that the amino precursor includes several heterocyclic components. The authors also describe E and Z selectivity and speculate on its origins.
Overall, not all arguments are solid but the work is publishable. However, writing should be improved as a lot of it is too verbose and vague. Additionally, all Russian should be translated into English. I give a few examples below.
Arrows in schemes are somewhat random. Sometimes authors forget to show enolate (e.g., the left part of Scheme 3) or have an arrow that points to empty space (from 6 in Scheme 3).
Explanation for the E-configuration for compounds 4d,e seems to be a circular argument that is also quite trivial (higher barrier).
Scheme 5 describes an interesting example where a hydrazine (which is accidentally incorrectly described as “hydrazone”) is unreactive while amine is. This is probably the most important result in the paper as it seems to provide another example of the recently discovered inverse a-effect (J. Amer. Chem. Soc., 2017, 139, 10799–10813).
Top sentence on page 7: it’s unclear which amino group authors are talking about.
-the authors proposed two different mechanisms for their transformation, both of which have a logical pathway. One can probably get some insight into which mechanism is more likely by looking at the reactions of 2 with 3 and 2 with 1 separately to compare how quickly the suggested intermediates 5 and 6 are formed. Seems to be an obvious control experiment.
DFT calculations could be very helpful to better understand the relative likelihood for such mechanisms. In addition, such calculations would allow for better understanding of the E to Z selectivity, which according to the authors has mainly to do with kinetic preference versus thermodynamic preference. There is no real proof for the later assumption.
-The schemes in the paper do not do a good job of supporting the text. Schemes should help the reader understand the topic and thus should illustrate discussions that are hard to follow. Schemes can also include compounds with cumbersome names since this paper has many such compounds mentioned.
As for the scope, although a range of amines was studied, there is no scope for the R group of furanone
Round 2
Reviewer 1 Report
Arrow pushing in schemes can still be improved. For example, where does arrow in compound 12 in Scheme 3 go? There should be an arrow that starts with an amine and goes to the carbonyl (to indicate the Michael addition) instead of going to a random atom in the ring. Please make sure that the schemes satisfy the standards of chemical literacy.
Otherwise, the paper is improved and can be published
Author Response
As an answer to comments of the Reviewer we have made chaniges to Scheme 3.
We also asked the professional English translator (Mr Dmitry Tychinin, IBPPM RAS) to correct the text of the manuscript.
Reviewer 2 Report
No additional comments.
Author Response
We have made changes into the manuscript according to comments from Reviewers